# Process Safety Management Quality in Industrial Corporation for Sustainable Development

Adam S. Markowski [1,*], Andrzej Krasławski [1,2], Tomaso Vairo [3] and Bruno Fabiano [3,*]

1 Department of Safety Engineering, Lodz University of Technology, 90-924 Lodz, Poland; Andrzej.Kraslawski@lut.fi
2 School of Industrial Engineering and Management, Lappeenranta-Lahti LUT University of Technology, 15101 Lahti, Finland
3 Civil, Chemical and Environmental Department (DICCA), Polytechnic School of Genoa University, 16126 Genoa, Italy; Tomaso.Vairo@edu.unige.it
* Correspondence: adam.markowski@p.lodz.pl (A.S.M.); brown@unige.it (B.F.)

**Abstract:** In recent years, also in connection with Covid-19 pandemics and enforced restrictions, there has been the formation of large industrial corporations gathering separate companies with similar, sometimes complementary production profiles. This evolving trend has brought usually positive economic effects; however, it has also created some integration problems that include the process safety management. The Texas City BP accident in 2005 and its tremendous human and economic losses underlined the obstacles in defining a well-structured corporation process safety management. The main causes of the above-mentioned accident were connected to an inadequate safety culture at the managerial level. Strong leadership and high standards of corporate governance are required to inspire correct safety behavior in the staff. The so-called soft skills become even more important in the Industry 4.0 arena, where the foundation of the whole system is based on an intelligent use and interpretation of data. The importance of this aspect is confirmed by several post-accidental analyses of past events. Although some research on this topic has been already done, it is worth it to dedicate some effort to identifying specific factors which influence the corporate process safety management quality, and, once identified, to assess them. This paper applies the concept of "lessons learnt" for the identification of organizational and managerial aspects worth consideration in process safety management. Based on accident and literature reviews and expert opinions, the aim is to identify the major contributing factors among leadership and safety culture, risk awareness, knowledge and competence, communication, and information and decision-making processes. To self-assess the level of commitment of the top leaders in process safety management, a checklist approach is proposed, combined with a quantitative, weighted evaluation based on the Relative Efficiency Indicator (REI). Positive value of REI may ensure the effectiveness of process safety management in major hazard industries and their appropriate adaptation to the corporation community. The proposed method, which is validated in an actual case study, underlines the importance of an appropriate education, and of a more careful selection of HSE managers.

**Keywords:** corporation; safety culture; safety management; Seveso establishment

## 1. Introduction

Recently, in many East European countries, large industrial corporations are bringing together many industrial plants with similar production processes. Under the pressure of an even more competitive marketplace, individual plants need to join their forces and create large industrial clusters, for producing positive economic effects related to integration and scale economy. However, the other side of the coin is the need to deal with organizational shortcomings, which include safety and risk management, and this facet can be particularly relevant in "Seveso" plants, where the evolution in regulatory thinking has progressively integrated traditional occupational safety with process safety. In Europe, since 1982, safety

approaches were integrated into the EU legislation, with the so-called Seveso Directives (Directive 82/501/EEC [1], Directive 96/82/EC [2], Directive 2012/18/EU [3]). Moreover, the business environment is becoming more and more dynamic and competitive, and frequent turnover in the staff ("job hopping") has deepened the problem. Plant corporations are facing the conundrum of increasing production and, at the same time, achieving higher safety and environmental standards. Most of the corporations include Upper Tier Plants, under the umbrella of the last amendment of the European legislation focusing on prevention and control of major chemical incidents, known as Seveso 3 Directive, which means that safety policy standards are high. It must be mentioned that, for the first time since the first Seveso directive issued in 1982, Seveso III explicitly mentions specific procedures for safety performance indicators and/or other relevant indicators, to be utilized for monitoring the performance of safety management systems [4]. Consequently, leaders need to be well prepared to deliver high-level results within this topic [5]. This paper addresses specific aspects of corporate safety management that need to be considered by corporate management in process safety. The reports on the major process accidents, the detailed analysis of some events as well as the results from Seveso inspections in several industrial plants [6] were used to identify the essential safety-related aspects at the managerial level. The analysis of the above-mentioned sources allows the identification of six fundamental factors, which may drive effective performance of top managers for developing a corporate community whose aim is to ensure process and personnel safety and sustainability. The most relevant factors are related to leadership and high standard safety culture, as well as additional attributes, such as awareness and process risk assessment, knowledge and competencies, proper communication and information, effective decision-making, and resilience. A rather simple check-list approach is developed, which may be used to calculate the Relative Efficiency Indicator (REI) and self-assess the corporate management commitment within the field of safety management. The framework is mainly addressed to board members responsible for operations and development, executive directors, and managers of plants within a corporation.

## 2. Identification of Factors Necessary to Effectively Manage Process Safety in a Corporation

Ensuring safety in Upper Tier Plants requires robust roots in risk assessment and safety management systems. Such systems have been defined in different technical guidelines, e.g., ISO 31000 [7] and ISO 45001 [8], or else OSHA [9] in USA. Additionally, it must be remarked that the concept of "risk-approach" is also integrated in the sector-specific quality management reference ISO 29001:2020 [10]. Safety management includes three main components based on the Deming management cycle:

1.  Designing the safety foundations of safety by delineating general, establishing safety principles and organizing the system by allocating authorizations and responsibilities.
2.  Delivering and mastering safety by developing and empowering appropriate management procedures.
3.  Checking and evaluating the system performance through audits and check-ups to double-check the attainment of goals adopted for the safety policy and introducing adjustments.

Management processes concern the so-called management components, which cover specific areas of industrial processes and safety management with strictly defined management procedures. The above-mentioned, normalized management systems have different structures when it comes to the type and number of elements. The PSM standard includes 12 components; the OSHA norm, 14 components; while some European companies covered by the Seveso Directive include 13–15 components [11].

The implementation and effectiveness of those processes are dependent on company resources, i.e., human resources, economic resources, knowledge and experience, other external circumstances, and regulations, as well as on multiple organizational factors. Several recent studies were performed on actual implementing and improving existing

SMS. It seems well worth mentioning Demichela et al. [12] who evidenced that risk analysis (RA) provides sizing criteria for the whole SMS and helps to define the objective of the management system itself. Bragatto et al. [13] outlined a novel framework based on the bowtie model to improve the practical implementation of SMS in small-sized enterprises, while in [14] it is evidenced the relevant role of managerial and organizational factors in developing risk analysis studies addressing risk-based decisions.

The main method used to identify organizational and culture-related factors, which are principal causes of accidents, consists in using historical accident and incident-related data. The need of a historical accident analysis is increasingly recognized in the industrial sector, to understand the triggering causes [15], avoiding the repetition of the same mistakes noticing critical aspects of the process that often go unnoticed at the design stage.

Historical data on industrial accidents are available on several following databases, e.g., FACTS currently managed by the Unified Industrial & Harbour Fire Department in Rotterdam-Rozenburg [16], eMARS [17], Process Safety Incident Data PSID [18] and several surveys on selected accident scenarios were developed using, for instance, the Major Hazards Incident Data System (MHIDAS) [19], or FACTS database [20]. In the following, we do not provide a thorough accident synopsis, nor do we list all the learnings and changes that came from the selected incidents, but we highlight the key issues related to safety management items focusing on accidents resulting from leadership lack and evidencing the need to strengthen safety management systems. Table 1 lists selected major accidents caused by safety management-related aspects.

**Table 1.** Major accidents and main root causes.

| Date | Location | Industry | Fatalities | Main/Root Causes | Ref. |
|---|---|---|---|---|---|
| 10 July 1976 | Seveso | Chemical | - | Human error, lack of process knowledge Emergency preparedness | [21] |
| 2 December 1984 | Bhopal | Chemical | 8000 immediately 12,000 thereafter | Process safety and ageing management system Emergency preparedness | [22–24] |
| 26 April 1986 | Chernobyl | Nuclear power plant | 985,000 | Human error in design Production pressure Absence of proof tests Leader error | [25] |
| 28 January 1986 | Challenger space shuttle | Space | 7 | Organization failure Pressure on success | [26] |
| 6 July 1988 | Piper Alpha Platform | Gas and oil | 167 | Management of change errors Production pressure | [27] |
| 3 October 1989 | Philips, Texas | Chemical | 23 | Human error | [28] |
| 13 May 2000 | Enschede, The Netherlands | Manufacturing | 22 | Lack of operational discipline | [29] |
| 21 September 2001 | Toulouse | Chemical | 30 | Lack of knowledge Poor hazard identification | [30,31] |
| 23 March 2005 | Texas City | Oil and gas | 15 | Failures in corporate management and culture | [32] |
| 20 April 2010 | Mexican Bay USA | Oil and gas | 11 | Lack of supervision | [33] |
| 17 April 2013 | West, Texas | Logistics | 15 | Lack of risk awareness | [34] |
| 12 August 2015 | Tianjin, China | Logistics | 173 | Failures in management system | [35] |

**Table 1.** *Cont.*

| Date | Location | Industry | Fatalities | Main/Root Causes | Ref. |
|---|---|---|---|---|---|
| 22 March 2018 | Kralupy, Czech Republic | Chemical Refinery | 6 | Human error and lack of supervision | [36] |
| 29 October 2018 10 March 2019 | Boeing 737 Indonesia Ethiopia | Air traffic | 181 157 | Design errors Production and profit pressure. Gaps in risk management | [37] |
| 4 August 2020 | Beirut port Lebanon | Storage | 204 | Lack of risk awareness Poor process safety Management | [38] |

Even if far from being complete, the above list suggests that, although over time new solutions in risk and safety management have become available, several issues, linked mainly with oversights and human errors in individual elements of safety management systems, constantly come back. Human errors are crucial, and they happen in the design or operational stage. Detailed knowledge about the root causes related to organizational and cultural factors is not so common during the forensic investigations after an accident. Such knowledge is sometimes available for accidents that caused severe consequences and triggered strong public pressure. Forensic investigation of the Chernobyl disaster for the first time addressed the issue of negative safety culture as the root cause of the nuclear catastrophe [24].

The most relevant analyses were performed as a follow-up of the explosion in Texas City in 2005 [32]. Table 2 summarizes the conclusions of the Baker Panel on corporate safety management [39], obtained after a thorough analysis of the accident immediate and root causes based also on detailed questionnaires.

**Table 2.** The Baker Panel conclusions on shortcomings in management factors.

| No. | Impact Factors |
|---|---|
| 1 | Absent or poor leadership of the corporate management in safety |
| 2 | Shortcomings, or rather negative safety culture and climate (infringing procedures, inability to learn, cost cuts and a system of awards related with it, weaknesses in the safety assessment resulting from compliance assessment, not risk assessment) |
| 3 | Inadequate organizational structure and unspecified scope of management competence and responsibility in the area of safety |
| 4 | Insufficient knowledge and experience of leaders and no support to production managers |
| 5 | Underestimated need to assess safety |
| 6 | Absence of monitoring and Board's supervision over advances made in process safety |
| 7 | Attention paid mainly to occupational safety and safety indicators (IIR) |

The above conclusions were confirmed by Hopkins [40] and are representative for some other process accidents, including an explosion in Tesoro Refinery (2010) and fire in Chevron Refinery in 2012 [41]. A research investigation performed on more than 30 high risk plants in Poland evidence similar issues, especially poor safety culture and lack of risk awareness [42].

In analyzing accidents statistics, there is no doubt that the leadership has a major impact on the effectiveness of safety and that PSM is recognized as the primary approach for establishing the level of safety in operations required to manage high-hazard processes and plants. Leadership requires many technical, social, and conceptual skills at the management level because it involves considering the corporation as a community that can ensure safety. Personal leadership skills supported with a solid system of communication and information are very helpful. Concerning the other aspects of safety culture, there are

misgivings around the competences of new management staff, the ability to generate a self-learning environment that takes advantage of historical data, the issue of "cost cutting", which typically hinders safety measures and budgets allocated to training and learning in the first place.

Risk awareness at each level of installation development, from its design through exploitation up to the decommissioning, is another important aspect and one of the root causes of many accidents. DuPont believes that risk awareness is the key to ensuring operational discipline; the latter is defined as an engagement and commitment of each member of an organization in order to correctly comply with her/his duties at any moment of time [43].

At the same time, operational discipline is actually reinforced by positive safety culture and leadership functions related to the authority and professional position.

Another element that testifies to the importance of risk awareness and communication is the number of warnings and penalties imposed by the OSHA, which placed the issue at the top of its statistics for 2017 [44].

All the above-mentioned safety management factors can work properly only when the decision-making system as well as communication and information flow operate properly.

## 3. Safety Management in a Corporation: Leaders' Responsibilities

Although the main focus of corporate top management is to define business strategic goals creating added value, drafting action plans, and supervising their implementation, a sustainable development program must also be central in the actions of corporate boards. Sustainability includes the protection of life and health, environmental safety, and resource integrity.

The Seveso III Directive stated that, within the context of relevant Union legislation, the Commission may examine the need to address the issue of financial responsibilities of operators in relation to major accidents, including issues related to insurance. Operators of certain industrial installations, whose operation is defined as a hazardous activity due to the presence of hazardous substances in certain quantities, are liable for the damage caused by an industrial accident. Operators are required to cover liability for this risk by financial security. In this regard, it is well worth underlining the direct leader of financial responsibility.

Effective sustainability management requires permanent involvement of corporate management. The system must be founded on safety principles, e.g., [45]:

1. All types of incidents and accidents can be prevented.
2. Legal, financial, and organizational responsibilities for safety fall on the management.
3. Safety is an inherent part of any process/task.
4. Safety performance is strongly dependent on staff knowledge and training.
5. Safety of each process/task must be assessed.
6. Any divergence from established practices and standards must be identified and immediately corrected.
7. All accidents and incidents must be analyzed.
8. Staff living and resting conditions can make a difference.
9. Accident prevention is good business.
10. Any program designed to ensure safety depends on people and their engagement.

Based on Section 2 we may formulate a general safety management model, which should be applied in the management of industrial corporations. The conceptual model outlined in Figure 1 includes six fundamental components for the delivery of programs and tasks in the area of safety management. Special importance is given to the central component, i.e., leadership and safety culture. Although not enumerated as a formal element within the safety management system in individual management standards, it is the "warp" holding the entire structure of safety management system. The remaining components are conceived on grounds of different amply applied elements of standardized management systems [7,8].

Additionally, we also need to consider external and internal circumstances and environment, in which a corporation operates.

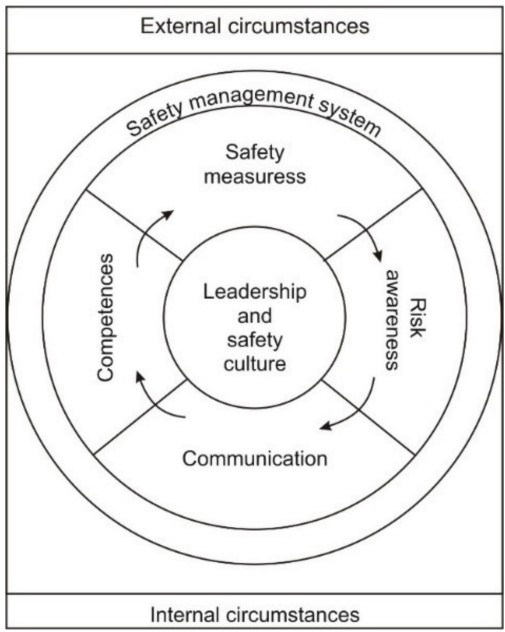

**Figure 1.** Safety management components at corporate top management level.

As a main limitation of the approach, it should be evidenced that the list of components is not exhaustive because it does not include other factors explicitly required in process safety management systems but addresses only those that are relevant at the corporate management level. A group of 15 experts from various countries, including both academics (i.e., PhD researchers and professors) and practitioners, (i.e., members of control authorities, facility managers, HSE consultant experts) was involved in a specific online survey in order to assess the quantitative weight of the performance factors. Their heterogeneous background allows covering the various relevant facets of the research matter thus ensuring a higher completeness of the results [46]. Based on historical data, the proposed estimates of the relevance of each of these aspects in the form of weight indicators (w) are summarized in Table 3.

It should be noted that a special attention is paid to leadership and safety culture, which exert decisive impact on corporate safety indicators [47,48].

The results of expert estimate, which was performed by an Analytic Hierarchy Process (AHP) [49], are reported in Table 4, where CR stands for consistency ratio.

The assigned weights (w) in Table 3 are derived from the AHP results.

**Table 3.** Importance of safety management components.

| No. | Component | Importance (in) | Value Assumed in the Analysis (in) |
|---|---|---|---|
| 1 | Leadership and safety culture (LSC) | 0.3–0.4 | 0.3 |
| 2 | Risk awareness (RA) | 0.2–0.3 | 0.2 |
| 3 | Communication and information flow (CI) | 0.1–0.2 | 0.1 |
| 4 | Skills and competencies (SC) | 0.2–0.3 | 0.1 |
| 5 | Action–decision-making process (A) | 0.2–0.3 | 0.2 |
| 6 | External and internal circumstances (C) | 0.1–0.2 | 0.1 |

**Table 4.** AHP-Group Result and Priorities of Individual Participants.

| Participant | LSC | RA | CI | SC | A | C | CRmax |
|---|---|---|---|---|---|---|---|
| Group result | 28.8% | 19.9% | 10.1% | 10.9% | 19.0% | 11.3% | 0.4% |
| expert15 | 20.0% | 20.0% | 10.0% | 20.0% | 20.0% | 10.0% | 0.0% |
| expert14 | 22.2% | 11.1% | 11.1% | 22.2% | 22.2% | 11.1% | 0.0% |
| expert13 | 31.8% | 13.8% | 5.4% | 20.9% | 20.1% | 7.9% | 5.4% |
| expert12 | 14.3% | 14.2% | 8.5% | 25.4% | 25.4% | 12.2% | 3.4% |
| expert11 | 15.6% | 17.7% | 7.7% | 17.0% | 31.9% | 10.2% | 5.1% |
| expert10 | 10.2% | 16.1% | 20.8% | 18.7% | 17.8% | 16.3% | 3.1% |
| expert9 | 11.1% | 11.3% | 8.1% | 27.8% | 29.8% | 12.0% | 3.4% |
| expert8 | 33.1% | 26.3% | 6.2% | 9.6% | 11.6% | 13.2% | 4.3% |
| expert7 | 32.5% | 18.9% | 10.0% | 14.1% | 16.6% | 7.9% | 2.2% |
| expert6 | 16.4% | 38.9% | 6.8% | 8.9% | 14.9% | 14.1% | 7.2% |
| expert5 | 26.8% | 26.8% | 8.1% | 13.3% | 16.9% | 8.1% | 1.2% |
| expert4 | 29.6% | 29.4% | 6.5% | 11.5% | 13.3% | 9.7% | 4.9% |
| expert3 | 21.1% | 30.1% | 6.9% | 9.2% | 17.4% | 15.2% | 3.5% |
| expert2 | 35.1% | 15.8% | 6.0% | 15.8% | 19.4% | 8.1% | 2.0% |
| expert1 | 30.8% | 18.5% | 9.6% | 16.9% | 15.3% | 8.9% | 1.6% |

### 3.1. Leadership and Safety Culture

Strong and engaged leadership, able to manage and influence the accomplishment of process safety tasks, is the first and first-ranking component of effective management. Leaders should be able to prioritize tasks in process safety management and encourage their staff to perform effectively. Taking active part in meetings is not enough; leaders must be visible in the company, interact with the staff to promote and discuss safety issues and continuously strive for improving safety. Staff operations, behavior and problem solving are just some further aspects to be faced by the management. The knowledge of the elements which helps in conducting an in-depth examination of the problem is relevant, e.g., HSE manual [50,51] proposes 7 such factors while 10 main items are considered in the Safety Culture Maturity Model (SCMM) [52]. DuPont [53] exhibits a very interesting approach, based on a set of checklists including 24 job categories related with only three components (leadership, structure, processes, and activities). Next, the set of questions is applied to four different positions (corporate management, management/supervision of a plant, operating staff, and a specialist). The approach allows for the assessment of the relative safety culture strength in each plant and therefore for undertaking the appropriate correction actions. One of the conclusions drawn from the safety culture assessment methodology applied to DuPont plants reveals a significant divergence in assessments made by the management and by the staff. For instance, the question whether safety is of primary importance was answered positively by 88% of managers and only by 51% of workers.

In order to develop a simplified model of practical applicability, we selected seven components apparently representative for this group of workers to assess the performance of leadership and safety culture at the level of corporate management and manufacturing plants. The assessment is split into four score categories, as shown in Table 5.

### 3.2. Risk Awareness

The issue of understanding the importance of process safety covers all of the life cycle of a given process, i.e., starts from the design, construction, exploitation, and decommissioning of an installation in all its strategic, operational aspects, maintenance, storage and logistics. Some aspects of risk are important at the design stage and others at the operational stage. Process safety results from a complete risk awareness because it is a personal characteristic, which should be exhibited by every manager and front-line worker. Risk awareness needs general technical knowledge supported with analytical knowledge in the area of risk management. Managers always perform an important, often determining, role in evaluating alternative technological, technical, and organizational projects in the

light of the risk of a severe accident and costs involved in it. Risk management is important for managers at operational stage, in which technological, technical, and organizational changes often occur and available operational procedures usually are unable to predict all possible divergences and consequences for risk. Understanding them by performing risk assessments allows taking adequate decisions and ensures flexible operational discipline. In order to ensure safety performances, each manager and worker must be aware of risks involved in the operational process and be attentive to the risk possibility. The set of check-up questions that can be used to assess the risk awareness is presented in Table 6.

**Table 5.** Assessment of basic components of "Safety culture and leadership".

| Component | Poor (1) | Fair (2) | Good (3) | Excellent (4) |
|---|---|---|---|---|
| Corporate safety policy | Lack | Exists with some deficiencies, e.g., lack of the statement on management responsibility for safety | Determined but not officially communicated in social media | Precisely determined and openly communicated |
| Corporate safety principles and standards | Lack or only some little known and applicable | Exist but are neither complete nor updated | Complete and updated | Fully complete, updated and well known in the company |
| Organization of safety and security services (structured) | Few experts with clear responsibilities for safety and security | Main focus on health and safety at work, not on process safety | Fit for purpose | Well determined and reporting directly to CEO |
| Leadership and involvement | Rare on-Board agenda | Only a few the relevant leadership features | but reporting to lower lever of management | Permanently present on board agenda |
| Safety aspects included into responsibilities | Lack of involvement and charisma | Scope of responsibilities covers safety aspects | Leaders directly involved in safety matters | |
| Visibility of personal engagement | Meaningless requirements | Manager sometimes seen in the plant | Recognition of responsibility for safety | Full authority, charisma, respect, and ability to convince the staff |
| Ability to adopt a holistic approach to safety | Sees only the front page | Recognizes more aspects | Always present at accidents | Ability to create a self-learning organization |

**Table 6.** Assessment of basic components of "Risk awareness".

| Component | Poor (1) | Fair (2) | Good (3) | Excellent (4) |
|---|---|---|---|---|
| Importance of hazard and risk analysis (H&RA) in design, operation and maintenance | No H&RA use (or minimal) in design and operation | H&RA is based on the result of compliance analysis only | Understanding and application of the H&RA methods in design and operation | Well understood and fully applied the most advanced methods of H&RA. |
| Sensitivity to the hazard and risk analysis (H&RA) | H&RA almost neglected. No attention paid to hazards addressed in procedures | Sensitivity is enforced by formal results of audits | Sensitivity to risk is a natural behavior of leaders. | Sensitivity to the hazard and risk analysis (H&RA) |
| Initializing and promoting H&RA among the staff | No attention | Only as required by regulations. Rare discussions on the importance of risk management. | Frequent discussions at Board meetings on the use of H&RA | Proactive approach to fully adhering to H&RA in processes and procedures |
| Learning from incidents/accidents/near misses | Not practiced at all or disregarded | Little used and based only on own experiences | Partial, using available accident databases in H&RA analyses | Fully practiced and used as a support in H&RA analyses, risk communication to the staff and budget decisions |
| Corporate management discussions on the probability of accident risk. | Rare | Only after an accident has happened within the company | Frequent, take account of historical data | Management meetings always begin with safety issues. |

### 3.3. Communication and Information Flow

As is amply known, management functions include facilitating the communication process. Managers should ensure full information flow for effective safety management drawing on information obtained from inspections and reviews, analyses of safety indicators, and conclusions from incident and near miss investigations [54,55]. An advantage of large industrial corporations lies in their versatility. Even when their production profile is similar, their installations differ with applied technologies, detailed technical solutions, experience of technical staff, often working under different climatic, geographical, legal, or even cultural circumstances. Proper knowledge management and information exchange within the corporation creates a competitive advantage in process safety. It is paramount to use and include the so-called historical data in the information flow, exchange of experiences within sectorial organizations and conclusions from management system audits to continuously improve the performance. The checklist included in Table 7 includes questions to self-assess the performance of communication and information system in the context of safety management.

**Table 7.** Assessment of basic components of "Communication and information".

| Component | Poor (1) | Fair (2) | Good (3) | Excellent (4) |
|---|---|---|---|---|
| IT system availability | Guarantees only limited connections within a plant | Information flow available in several plants | All possible means of communication are available, including conclusions from the meetings of the Safety Committee and the Board of the corporation | Like in "Good" plus information from social media, conferences and technical literature |
| Formal and informal flow of information within a corporation | Missing or very much limited within one plant | Limited to some plants, not all | Available to all plants with difficulties in collecting responses from the superiors | Full interpersonal and inter-plant contact, communication and information exchange |
| Availability of data including historical data | Unavailable or hard to find | Available only to selected leaders within a given plant | Broad access to all leaders, including global data on accidents | Very good access, together with general safety indicators and their trends |
| Protection of the communication system | No special safety measures | Only physical protection and limited cyber security system | Integration of various protection systems and methods, e.g., LAN and WAN separated from the Internet | Fully integrated system of global protection of communication and information system |
| Communication system under the threat of severe accident | Depends on emergency power supply | Some lines supplied by the UPS | Advanced systems of emergency power supply | Fully reliable systems of emergency power supply. Communication systems providing information about emergency situation to the public |
| Resilience to fake information | Lack: no critical thinking and each info may enter to communication system | Just very limited | Adequate: critical thinking to separate fake news from real one | Full formal analysis applied with critical thinking |

### 3.4. Skills and Competence

Leaders must guarantee that competent managers, engineers, and other auxiliary staff at any plant/installation can cope with process and technical hazards. Staff skills in risk analyzing and assessment need to be enhanced through training courses and postgraduate studies. Leaders should also be able to communicate risk not only to their workers but also to society, to local community, under normal and emergency conditions. The stability of process and technical hazards staff is also important. Table 8 provides a list of basic questions used to assess the importance of skills and competences in safety management.

**Table 8.** Assessment of basic components of "Skills and competencies".

| Component | Poor (1) | Fair (2) | Good (3) | Excellent (4) |
|---|---|---|---|---|
| Specify the scope of competencies in the area of safety knowledge | Not specify or little knowledge in the field of safety | Basic engineering knowledge | Basic engineering knowledge and safety basics | Complete and confirmed knowledge in the field of engineering and safety |
| Understanding the importance and meaning of safety management SMS | Little | Formal resulting from legal requirements | Good understanding of the importance of SMS | Very good, demonstrated at Board meetings |
| Taking care of safety training and education as an element of human resource development | Little attention, the management believes it is individual responsibility | Observing only the scope required by formal regulations | Usually, positive response to initiatives of skill improvement | There is a plan of human resource development designed to improve competencies of all operational and maintenance staff |
| Management competencies in decision making | Little consideration given to safety aspects | Decision making focused exclusively on the "cost of safety" | Decision making based on "cost and benefit" analysis | Support to holistic approach and ability to measure safety efficiency |

## 3.5. Action

Action is the main attribute of leadership, and, for the management, it includes the decision-making process. Decisions are made mainly by executive directors and top managers. The drafting of adequate plans aiming at avoiding accidents is a management duty, as well as ensuring maintenance and proper equipment. Equipment needed for process safety should be included in investment plans as well as in merger and acquisition plans for enterprises within an industrial corporation. Table 9 includes the set of basic questions that can be helpful in assessing the efficiency of leaders' actions in the field of safety.

**Table 9.** Assessment of basic components of "Decision-making process".

| Component | Poor (1) | Fair (2) | Good (3) | Excellent (4) |
|---|---|---|---|---|
| Drafting safety improvement plans | Does not exist or exists to a very limited extent | Only basic programs for health and safety in selected production units | An integrated improvement program for most selected installations | An integrated continuous improvement program for all plants |
| Systematic review of safety improvement plans | Does not exist or is very limited | Only for some plants within the corporation | For most plants, not very regularly | For all plants, regularly, information disseminated to all |
| Safety Committee activity | Does not exist | Only for special tasks, e.g., for explosive zones | For most areas that need support | All Committees have been set up, their remit is defined, and they work on a regular basis |
| Reviewing conclusions and expenditure resulting from safety audits | Does not exist | Rare | Frequent | Regular |
| Problem solving method | Intuitive, doing it "our" way is always the best | Partial use of problem analysis principles | Using problem analysis principles combined partly with alternative solutions | Rational analysis considering alternatives and the selection of the best solutions for implementation |

## 3.6. Boundary Conditions

Boundary conditions are limitations imposed by circumstances that affect the decision-making process at the management level. These circumstances can be divided into external and internal. Each plant is subject to external competition, owners' and client's requirements, and the pressure of local community and administrative authorities, which impose specific legal and administrative requirements. Good relations with the external environment are crucial for effective corporate management. External factors decide on how

much resources are allocated for safety, safety priorities, and standards applied within a given corporation. This is true of any development stage of a given project, starting from the design stage through construction, exploitation, and decommissioning of industrial installations. As the history of accidents and catastrophes teaches us, often the pressure to increase production and insufficient funding forces corporations to compromise on safety standards, leading to serious losses, e.g., [22–32].

Internal circumstances include the specificity of a given plant, which exhibits itself mainly in cultural factors. The process is assisted by good organizational production structure, prevention and maintenance services, as well as by a proper promotion and career planning system. Table 10 summarizes a set of basic questions that can be helpful in assessing the efficiency of leaders' performance in connection of safety performances.

**Table 10.** Assessment of basic components of "external and internal circumstances".

| Component | Poor (1) | Fair (2) | Good (3) | Excellent (4) |
|---|---|---|---|---|
| Compliance with formal requirements | Rarely | Only for selected tasks | Almost always with minor exceptions | Always compliant with updated requirements |
| Impact of competition and production goals | Production over safety. "Do more for less" principle | Production over safety with attention to safety aspects | Safe production is ensured | Well balanced safety vs. production |
| Local environment impact | Hostile | Separate safety goals, no conflicts | Correct relations with social environment | Good relations and shared interests |
| Resilience to ownership relations and decisions | No personal stance. Diverse goals | Ownership decisions even for divergent views are usually approved | Correct partnership relations with owners | Good understanding and full consent |
| Leaders and staff compliance with principles, norms, standards, and behaviors | Big differences | Small differences | Correct | Full and above average |

*3.7. Calculating Relative Efficiency Indicator WWS [REI]*

To assess performance efficiency and the involvement of leaders in safety management a relative efficiency indicator WWS has been proposed, with a 4-point Likert scale to measure the company effectiveness as discussed briefly in Table 11. Values of the WWS have been estimated by experts and they will be used in relative comparisons and assessments of deviations from maximum values with $WWS_{max} = 4$ and $WWS_{min} = 1$.

**Table 11.** Categories of the WWS indicator.

| WWS Degree | Indicator | | REI [-] |
|---|---|---|---|
| 1-Low | "Negative". Reactive, based on natural instincts | Needs immediate correction | <1.5 |
| 2-Medium | "Developing" | Reactive, needs correction over a longer time horizon | ≥1.5 |
| 3-Good | "Best practice". Proactive, based on many individual initiatives | Needs small corrections, uses cost and benefit analysis | >2.5 |
| 4-Excellent | "Continuous improvement". Proactive–collaboration and involvement of all the staff | Does not need improvements | >3.5 |

WWS indicators are determined separately for each component of the model and subsequently the overall relative efficiency indicator (WWS) is estimated on the basis of Equation (1):

$$WWS = (WWS_{KBP} \times w_1 + WWS_{SR} \times w_2 + WWS_{KI} \times w_3 + WWS_K \times w_4 + WWS_D \times w_5 + WWS_W \times w_6) \quad (1)$$

where $WWS_{KBP}$, $WWS_{SR}$, $WWS_{KI}$, $WWS_K$, $WWS_D$, $WWS_W$ represent values of WWS indicator for each component determined based on the checklists from Tables 2–7, and $w_1$, $w_2$, $w_3$, $w_4$, $w_5$, $w_6$ are weighted indicators for individual components specified in Table 1.

Although each indicator is assessed against full categories (from 1 to 4), in practice, fractional values are also admissible (e.g., 2.5). At this stage of the work, no attempts were made to separately evaluate the weight of each component from the checklists, which means they are considered as equally valuable.

Results of calculations can be presented using a radar chart, as exemplified in Figure 2.

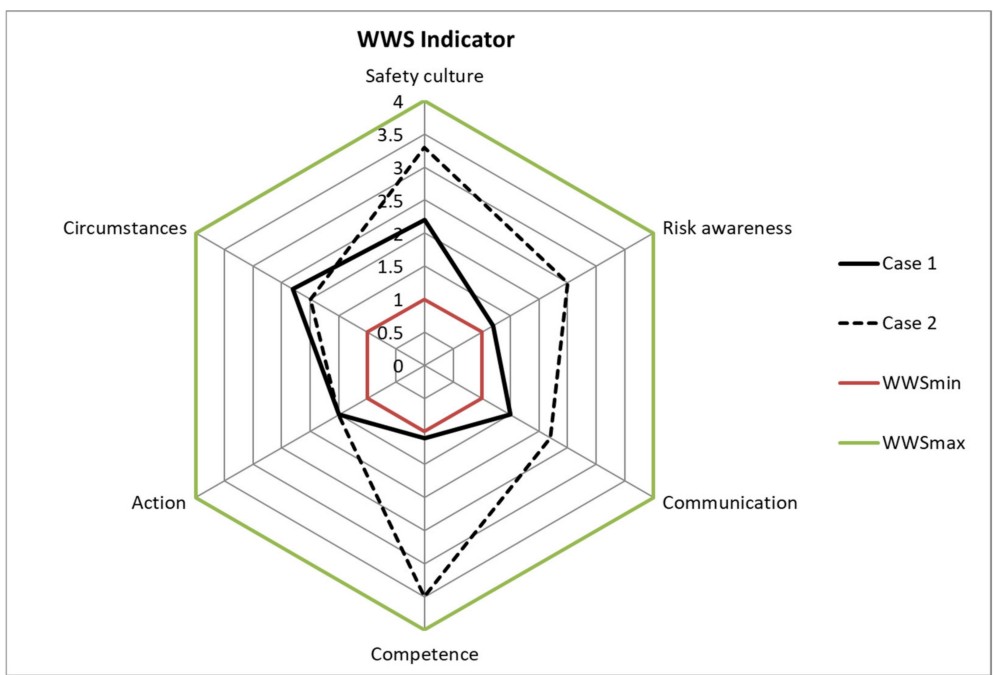

**Figure 2.** Radar chart for the identification of WWS profiles.

The chart presents maximum and minimum values of the indicator, i.e., $WWS_{max}$ and $WWS_{min}$. For instance, WWS values are given for two practical examples (1 and 2), which helped identify components in which shortcomings and differences are still significant. It allows putting adequate correction measures in place. Notably, within a continuously improving system, the threshold criteria for these indicators may shift towards more stringent values.

## 4. Applicative Case-Studies

In order to test the applicability of the proposed methodology, we tested the application into three Upper Tier Seveso Installation in Northern Italy, with a well-established corporate safety management system, namely:

-    A petrochemical storage facility (case 1–blue color);
-    An LNG terminal (case 2–orange color);
-    A process plant (case 3–grey color).

The results of the proposed checklist were subsequently compared with the results of planned institutional audits on the SMS of the Seveso sites, as summarized in Tables 12–14 for the three audited facilities. This kind of comparison— presented for the first time, at least to our knowledge—may evidence the actual strengths and drawbacks of the developed simplified tool. Such an approach may allow going beyond the standard assessment to confirm that everything was complying with the best industrial practices: results obtained by internal audits and trying to identify the weaknesses and the most relevant safety-related issues experienced by the facilities are presented in Figure 3.

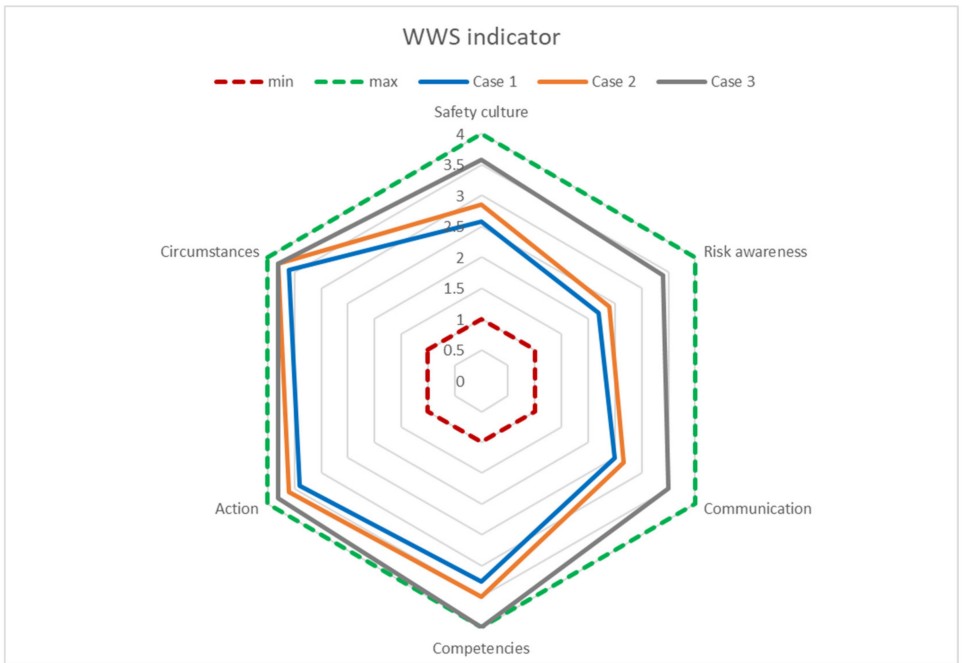

**Figure 3.** Radar chart for the identification of WWS profiles in the three case studies.

*4.1. Checklist Results*

**Table 12.** Checklist results for Case 1.

| CASE 1 | | | | |
| --- | --- | --- | --- | --- |
| **Safety Culture** | **Risk Awareness** | **Communication** | **Competencies** | **Action** |
| 2 | 3 | 2 | 3 | 4 |
| 2 | 2 | 3 | 3 | 3 |
| 3 | 2 | 2 | 4 | 3 |
| 3 | 2 | 2 | 3 | 4 |
| 3 | 2 | 3 | | 3 |
| 2 | | 3 | | |
| 3 | | | | |

**Table 13.** Checklist results for Case 2.

| CASE 2 | | | | |
| --- | --- | --- | --- | --- |
| **Safety Culture** | **Risk Awareness** | **Communication** | **Competencies** | **Action** |
| 3 | 3 | 2 | 3 | 4 |
| 4 | 2 | 3 | 4 | 4 |
| 3 | 3 | 2 | 4 | 4 |
| 3 | 2 | 3 | 3 | 3 |
| 3 | 2 | 3 | | 3 |
| 2 | | 3 | | |
| 2 | | | | |

**Table 14.** Checklist results for Case 3.

| CASE 3 | | | | |
| --- | --- | --- | --- | --- |
| **Safety Culture** | **Risk Awareness** | **Communication** | **Competencies** | **Action** |
| 4 | 3 | 3 | 4 | 4 |
| 4 | 4 | 4 | 4 | 4 |
| 4 | 3 | 4 | 4 | 4 |
| 3 | 4 | 4 | 4 | 4 |
| 4 | 3 | 3 | | 3 |
| 3 | | 3 | | |
| 3 | | | | |

*4.2. Institutional SMS Audits*

The Italian situation on the global results of the SMS audits over the time span 2019–2020 is depicted in Figure 4. Inspection scheduling is usually performed in the oil and gas industry according to standards implementing quantitative risk-based techniques, e.g., [56]. As recently reported [57], cause accident analysis evidences the crucial role between risk controls and the Safety Management System also in activities like hazardous material transport via pipeline not included under the umbrella of Seveso Directives aiming at preventing major accidents at industrial facilities. Figure 5; Figure 6 respectively illustrate the major and minor non-compliances resulting from inspections in the given plants.

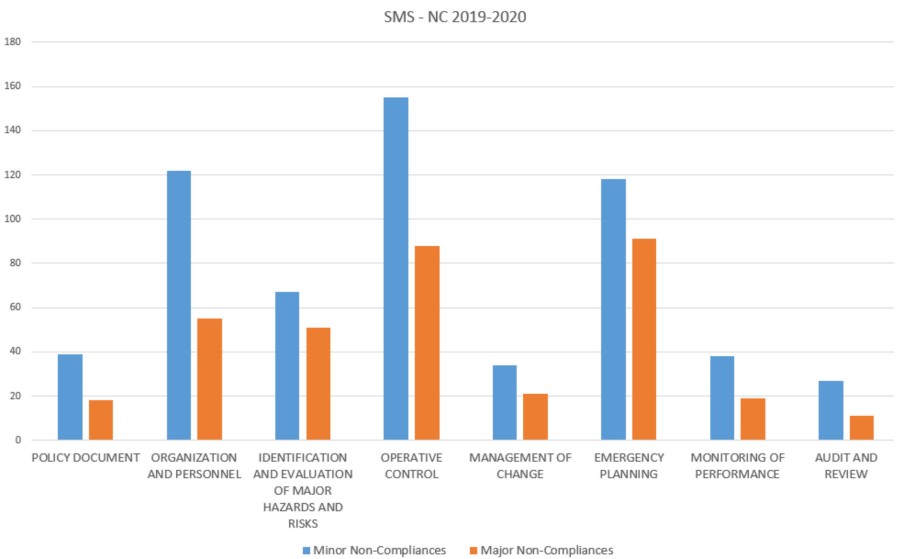

**Figure 4.** SMS audits bar chart for 2019–2020 results (Italian Ministry of Environment).

Concerning the three explored case studies, the actual results obtained from official routine inspection activity are summarized in Figures 5 and 6 respectively for major and minor noncompliances.

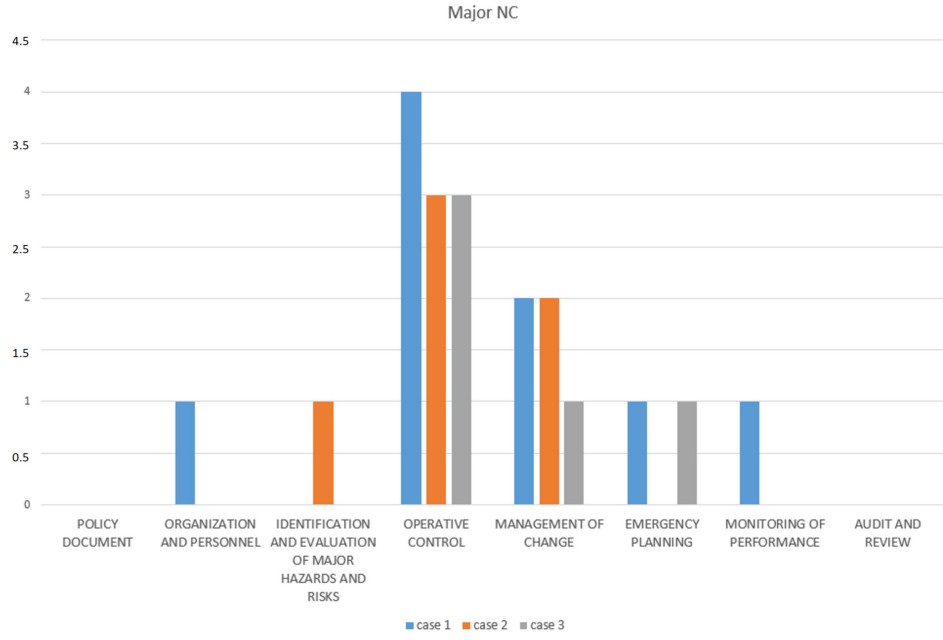

**Figure 5.** Major non compliances in the three case studies.

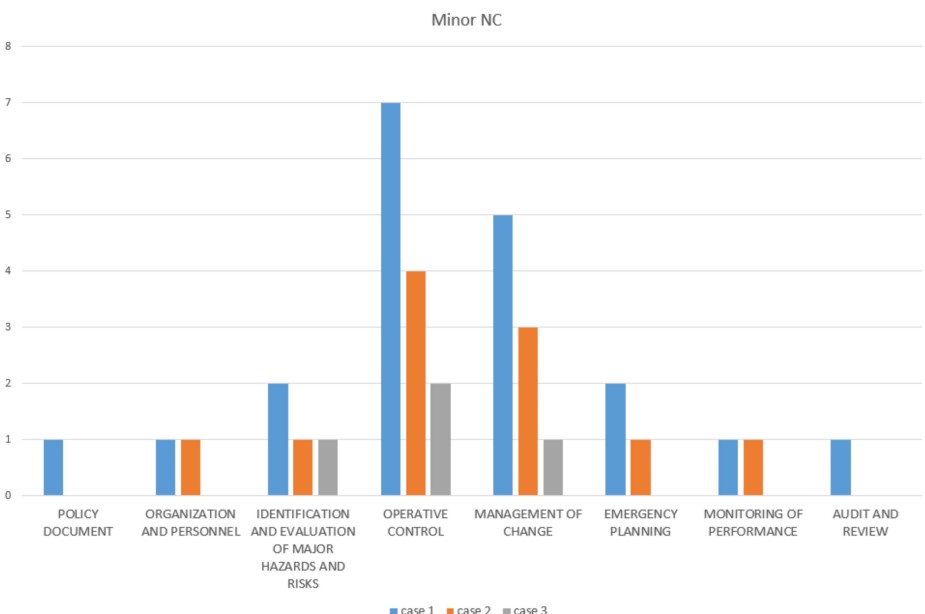

**Figure 6.** Minor non compliances in the three case studies.

### 4.3. Key results from Comparative Assessment

A statistical overview of the minor and major non-compliances, verified during the Seveso site inspections, points out that the most relevant critical point of the safety management systems turned out to relate to operational control, critical elements identification, maintenance processes management, ageing management and operational experience analysis.

Even considering its inherent limitations, the semi-quantitative analysis performed allows evidencing criticalities and highlighting the most vulnerable items of SMS. The weaknesses coming out from the checklist application in the three case studies are mainly related to risk awareness and communication.

The comparison of those results is not surprising at all.

In fact, a well-established operative experience relies on three main aspects:

- a good reporting system of near-miss, accidents, incidents
- a solid risk awareness
- an open communication between operators and management.

The operative control is deeply rooted in a reliable operational experience analysis, and the operational experience is based on an open communication between operators, which should not be blamed for the event reports, and the management, which should be aware of the importance of events identification and related hazards. The proposed work represents a structured approach capable of analyzing the firm safety management model and may help in identifying and prioritizing best improvement measures according to a transparent systematic process. The tool may help in pursuing continuous improvement in process safety by developing, using and acting upon effective leading indicators. In fact, to gain sustainable, continuous improvement, there is no alternative to reducing/eliminating at-risk human behaviors and improving safety culture [58]. Additionally, as recently argued by Pasman [59], an adaptation of the organizational structure would be required in many cases in order to gain a better appreciation of process safety, in addition to the many other priorities of top management team.

### 5. Conclusions

Historical data on serious accidents underline the central role of corporate management for safety performance. Well-established corporate safety management helps in building trust in a shared and communicative environment, inducing a collaborative work-

place, supporting sustainable growth, financial stability, and business integrity of any industrial corporation. A vital role can and should be played by the management whose knowledge, experience, and leadership traits (charisma, engagement, and commitment) exert the biggest impact on financial success of an organization. Nowadays, special tasks in this field are related to digitalization of management processes, including the application of new process technologies moving towards Safety 4.0.

This paper demonstrates how industrial process installations that use hazardous chemical substances can self-assess the main management and organizational factors that affect their safety standards. The assessment process concerns the six principal factors that may guarantee effective performance of top managers in a corporation. Due to the novel framework, this study only represents a first exploration of the topic, and this limitation must be considered when quantitative data are of interest. In fact, for example, the uncertainty of expert evaluation of relative weights is not fully discussed. The proposed checklist system, upon further validation, can provide, for each factor, suitable data for the effective calculation of the relative efficiency indicator. In perspective, results can help in self-assessing the engagement and efficiency of corporate management within the broad safety management domain.

The scope of the presented approach can be broadened to include also industrial sectors of activity different from the installations falling by law under the umbrella of the Seveso directives, which usually are already characterized by a high level of awareness on these issues. The outlined approach is not the only way of assessing leaders' engagement in safety management, but it can represent a relatively simple instrument for assessing the role of leaders in building the safety culture and thus contributing to sustainable, continuous improvement minimizing at-risk human behaviors. We may also assume that other positions, in the corporate organizational structure, may have different sets of factors and attributes, tailored to their respective specificities. The proposed method should encourage the young corporation to a more careful selection and to a more specific attention to the education and preparation of the senior HSE staff, inspiring as well further research and field validation in these directions.

**Author Contributions:** Conceptualization, A.S.M. and B.F.; methodology, A.S.M. and A.K.; validation, T.V., A.K.; data curation, T.V., A.S.M.; writing—original draft preparation. A.S.M.; writing—review and editing, A.K., T.V.; supervision, A.S.M., B.F. All authors have read and agreed to the published version of the manuscript.

**Funding:** This research received no external funding.

**Institutional Review Board Statement:** Not applicable.

**Informed Consent Statement:** Not applicable.

**Data Availability Statement:** Not applicable.

**Conflicts of Interest:** The authors declare no conflict of interest.

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
