# Peer review of "Process Safety Management Quality in Industrial Corporation for Sustainable Development"

_sustainability, doi:10.3390/su13169001_

Round 1

Reviewer 1 Report

The reviewed article presents an interesting research work focused on process safety management in industrial corporations. 

This article is suitable for publishing in this journal, but it requires some minor improvements. My comments and suggestions are as follows:

The authors refer to lines 77-78 "technical guidelines, eg, ISO 31000 [7] and ISO 45001 [8], or else OSHA [9] in the USA". There are also several types of risk: technical, financial, legal, health, managerial, etc. Since 2015, after the consolidation of management system standards, "risk-thinking" has already been integrated into all revised standards. For example ISO 29001:2020, Annex C: Risk and opportunity management and conformity assessment processes.

How many experts estimate data of the relevance of aspects? What method was applied for weight indicators (w), line/row 202?

In some countries, there is a crime/legal and/or financial responsibility of top management in industrial plants if their negligence causes an accident, damage to health, death or environmental damage, industrial plant damage. Could the authors of the article consider this relation in the next section, 3.8?

I recommend publishing this article after the implementation of my comments.

Author Response

The reviewed article presents an interesting research work focused on process safety management in industrial corporations. 

This article is suitable for publishing in this journal, but it requires some minor improvements.

Many thanks for your appreciation of our work.

My comments and suggestions are as follows:

The authors refer to lines 77-78 "technical guidelines, eg, ISO 31000 [7] and ISO 45001 [8], or else OSHA [9] in the USA". There are also several types of risk: technical, financial, legal, health, managerial, etc. Since 2015, after the consolidation of management system standards, "risk-thinking" has already been integrated into all revised standards. For example ISO 29001:2020, Annex C: Risk and opportunity management and conformity assessment processes.

Proper references are added and commented, following the useful suggestion.

How many experts estimate data of the relevance of aspects? What method was applied for weight indicators (w), line/row 202?

The answer is inserted in the revised paper with proper citation to the methodology. It is also added the result of expert elicitation performed by AHP, from which the weights are defined. The limitations of this method and of the study are also discussed in sections 4 and 5.

In some countries, there is a crime/legal and/or financial responsibility of top management in industrial plants if their negligence causes an accident, damage to health, death or environmental damage, industrial plant damage. Could the authors of the article consider this relation in the next section, 3.8?

The paper deals with Corporation SMS, and so the different approaches in different countries is implicit. What the authors wanted to point out is the involvement and efficiency of the top management of corporations (dealing with different countries) in safety related issues. The legal / financial responsibility is included in chapter 3. A relevant statement about financial responsibility in Seveso III Directive is inserted. to adequately underline this issue.

I recommend publishing this article after the implementation of my comments.

Many thanks for the useful comments, certainly improving the quality of the manuscript and addressing future research insights on the topic.

Reviewer 2 Report

Title: “Process Safety Management Quality in Industrial Corporation for Sustainable Development”

Manuscript number: SUSTAINABILITY_1298562

Author(s): Adam S. Markowski 1*, Andrzej Krasławski 2 , Tomaso Vairo 3 and Bruno Fabiano

Comments:

Overall, the topic covered of the manuscript is interesting and it may prove the way forward in addressing safety related issues in such industries. However, the manuscript may benefit if the authors address the following:

  1. Section 3, line 200-201, the authors argue that “…, we may propose expert estimates…”. However, it is unclear who were these experts. The associated weights given by the experts may be biased due to the background of these. For example, if the weights are given by ‘academics’ and not by safety professionals, or executives, etc., surely the results may not be representative. This review considers that this needs to be explained in some detail in the manuscript than currently is.
  2. What are the limitations of the study, if any?

Author Response

Overall, the topic covered of the manuscript is interesting and it may prove the way forward in addressing safety related issues in such industries. However, the manuscript may benefit if the authors address the following:

  1. Section 3, line 200-201, the authors argue that “…, we may propose expert estimates…”. However, it is unclear who were these experts. The associated weights given by the experts may be biased due to the background of these. For example, if the weights are given by ‘academics’ and not by safety professionals, or executives, etc., surely the results may not be representative. This review considers that this needs to be explained in some detail in the manuscript than currently is.

The full answer is inserted in the paper, including all pertinent details on expert selection, number and the statistical result by the AHP method, including reference studies

  1. What are the limitations of the study, if any?

The methodology can be broadened in scope, to include sectors of activity different than high risk industry, under the umbrella of the Seveso directives, which already show a higher level of awareness on these issues, and in the validation of the results, by applying it in different regulatory contexts. The actual limitations are properly commented in the text in sections 4 and 5, specifying as well possible lines for future improvements.

Many thanks  for encouraging words and the time spent for reviewing and practical suggestions, allowing us to improve the paper.

Reviewer 3 Report

Dear authors, 

thanks a lot for giving me the possibility to review your paper. I liked it. 

My considerations are : 

about literature : you cite Bhopal . Look at this https://doi.org/10.3390/app10030903 and add it to literature

about your investigation the question is : How can we do these parameters (REI etc. ) general and objective for all areas? How can we investigate? What is the methodological approach? Explain better. 

Author Response

My considerations are : 

about literature : you cite Bhopal . Look at this https://doi.org/10.3390/app10030903 and add it to literature

Proper reference was added; many thanks for the useful observation, raising the interest for the reader.

about your investigation the question is : How can we do these parameters (REI etc. ) general and objective for all areas? How can we investigate? What is the methodological approach? Explain better. 

The investigation was made through expert elicitation. As for comments by other reviewers, the methodology was detailes, adding as well the results of expert elicitation performed by AHP, from which the weights are defined. The limitations of this method are also discussed in sections 4 and 5.

Many thanks indeed for the accurate review and useful comments to improve the quality of the paper.

Round 2

Reviewer 3 Report

I find it complete, 

congratulations!